# GRADIENT-BASED DYNAMIC SPARSE TRAINING WITH ADAPTIVE REWINDING

## ABSTRACT

Deep neural networks (DNNs) deliver state-of-the-art performance across domains but impose prohibitive computational and memory costs. Pruning mitigates this challenge by removing unimportant parameters, yet conventional post-training pruning and reset-to-initial sparse training approaches incur high retraining costs or degrade performance on large models. To improve stability, prior post-training work suggests rewinding weights to intermediate checkpoints, though at the expense of costly offline analysis. We propose GDSTAR, a Gradient-based Dynamic Sparse Training framework with Adaptive Rewinding that supports models of different sizes and complexities without offline retraining. During training, GDSTAR (1) dynamically identifies stable rewind points using the Frobenius norm of gradients, (2) selects weights for pruning using accumulated gradient magnitudes, and (3) ensures stable optimization using a controlled pruning rate with exponential decay. Experiments across diverse DNNs and datasets show the efficiency and scalability of GDSTAR, which achieves up to 96% sparsity while maintaining accuracy, with only a 0.94% average drop compared to dense models. Compared to the state-of-the-art sparse training approach, GDSTAR improves accuracy by an average of 0.72% (up to 2.13%), under the same sparsity ratios.

## 1 INTRODUCTION

Deep neural networks (DNNs) learn complex patterns from large amounts of data (Goodfellow et al., 2016) at the cost of huge memory and computational demands (Sze et al., 2017). To mitigate these costs and enable more efficient deployment, model compression approaches, such as pruning and quantization, are widely used (Sze et al., 2017; Cheng et al., 2017; Chen et al., 2016; Courbariaux et al., 2016; Peste et al., 2021). Pruning directly targets the network's structure by removing unimportant parameters, especially weight elements and connections, thereby making the model sparser and more efficient in terms of memory and computation (Blalock et al., 2020). Early approaches typically perform pruning after training (Blalock et al., 2020; Lee et al., 2019; Molchanov et al., 2017b), followed by multiple rounds of retraining or fine-tuning to recover lost accuracy (Frankle & Carbin, 2018), which incur substantial overheads. In contrast, more recent approaches integrate pruning into the training phase itself, a.k.a., sparse training, offering a more cost-effective solution (McDanel et al., 2022; Golub et al., 2020; Yang et al., 2020; 2022; Lei et al., 2024).

Sparse training approaches, such as Procrustes (Yang et al., 2020) and DropBack (Golub et al., 2020), reset pruned weights to their initial values. They are effective for small-scale models, but are suboptimal for deeper or more complex ones (Frankle et al., 2020a) where the initial weights may no longer be well-aligned with the current optimization (Frankle et al., 2020b). The increased complexity and dimensionality of the loss landscape in large models cause this suboptimality by making the optimization trajectory more sensitive to stochastic gradient descent (SGD) noise early in training (Frankle et al., 2020a). The Lottery Ticket Hypothesis (LTH) (Frankle & Carbin, 2018) focuses on large and complex models by rewinding the weights to an intermediate point in training, known as the rewind point, which is typically defined by a specific epoch or iteration (Frankle et al., 2020a). This technique has been shown to enhance both training stability and final model accuracy. (Frankle et al., 2020a;b) However, LTH relies on an offline process to identify the appropriate rewind point, which involves fully training the model multiple times, making it computationally expensive (Renda et al., 2020). Consequently, none of the existing methods simultaneously supports integrated pruning into training and adaptive weight recovery for large-scale models.

We analyze rewind points in various model complexities and study how different attributes of a model are effective in determining the rewind point. To achieve this, we conduct experiments that track gradient statistics during training and utilize gradient noise to adaptively identify optimal rewind points without full retraining. We make three key observations. (1) The gradient is an

important factor in determining the rewind point as an alternative to LTH's costly offline analysis. In particular, the point during training where gradient noise stabilizes reliably marks an effective rewind point. (2) The accumulated gradient magnitude is a practical criterion for weight selection during sparse training, because gradients reflect each weight's contribution to the loss over time. (3) The rate of weight selection and the rate of weight reduction at each epoch are crucial, as preserving the network's scaffolding while dynamically increasing pruning in early epochs effectively improves performance and significantly cuts compute costs.

We propose a Gradient-based Dynamic Sparse Training framework with Adaptive Rewinding for models with various complexities, called GDSTAR. It adapts to model complexity by dynamically determining three key parameters: the rewind point, the selected weights, and the weight removal rates. Designed for compatibility, GDSTAR can be integrated into deep learning frameworks alongside other compression techniques without requiring modifications to standard training pipelines.

GDSTAR determines the rewind point dynamically during training by tracking the Frobenius norm of gradients, which quantifies the overall magnitude of gradient updates. At each epoch, this norm is computed, and the rewind point is updated whenever the norm reaches a minimum, indicating that stable learning dynamics (Frankle et al., 2020a) have been achieved. The weights from this epoch are then used for rewinding, ensuring that performance (e.g., model accuracy) is maintained. GDSTAR selects weights to be pruned based on the accumulated gradient magnitudes of weights. These gradients are divided into chunks to compute averaged quantiles, which reduces the computational cost of sorting. The number of weights selected for pruning at each epoch is determined by a dynamic pruning rate that helps the model adapt and maintain performance during training. Selected weights are not set exactly to the rewinding weights, but an exponential decay gradually reduces them toward zero, and a pruning threshold eliminates those already near zero.

We evaluate GDSTAR across a wide range of DNNs, including those already optimized for efficiency, on two datasets in terms of sparsity and accuracy (see § 5 for details on the DNNs and datasets). GDSTAR achieves an average sparsity of 93.64% and up to 96% sparsity with an average accuracy drop of only 0.94%, comparable to dense DNNs. Under the same sparsity ratios, the accuracy of GDSTAR consistently outperforms the state-of-the-art sparse training approach (Yang et al., 2020), with an average of 0.72% and a maximum of 2.13%. This paper makes these contributions:

- We study the shortcomings of sparse training and their reset-to-initial-weights strategies across DNNs with varying complexities.
- We propose an adaptive rewinding approach based on the Frobenius norm of gradients to address LTH's limitations and complexity in rewind point identification during training.
- We introduce GDSTAR, a sparse training framework that selects weights based on gradient accumulation and prunes them with a dynamic rewind-decay process.
- We evaluate GDSTAR, which achieves 93.64% average sparsity with an accuracy drop of 0.94%, compared to dense DNNs. Compared to state-of-the-art sparse training, it achieves up to a 2.13% accuracy gain under the same sparsity level.

## 2 BACKGROUND & MOTIVATION

DNNs consist of multiple interconnected layers, ranging from basic (e.g., convolutional and fully-connected) to more complex modules that are composed of the basic ones (Goodfellow et al., 2016). These layers are trained together to minimize a task-specific loss function through iterative gradient-based optimization, which allows them to learn hierarchical representations of the input data (Goodfellow et al., 2016). As shown in Algorithm 1 (see Appendix), during each training step, forward propagation produces predictions, which are compared with ground-truth labels to compute the loss. Gradients are then calculated through backpropagation (Goodfellow et al., 2016) and used to update the weights (filters or kernels). Once trained, the model can perform inference on unseen inputs.

Training highly parametrized DNNs needs excessive computation and memory requirements (Sze et al., 2017). Pruning has emerged as a widely used approach to reduce the size and complexity of DNNs by eliminating redundant or non-essential components such as weights, gradients, and attention heads (LeCun et al., 1990; Li et al., 2017; Hoefler et al., 2021).

Table 1 summarizes representative pruning approaches, highlighting their pruning targets, criteria, and integration with the training pipeline. Most existing approaches operate in the post-training phase. They first train the model to convergence, then prune it and fine-tune to recover lost accuracy (Lee et al., 2019; Frankle & Carbin, 2018; Han et al., 2015; He et al., 2018; Molchanov et al., 2017b). Automated Model Compression (AMC) (He et al., 2018) prunes the entire channel after training, focusing on hardware-aware compression to balance accuracy and efficiency (He et al.,

Table 1: Comparison of Pruning Techniques

| Method | Target | Criteria | Stage | Inference | Rewind | $k^*$ Selection | Decay Strategy |
|---|---|---|---|---|---|---|---|
| AMC | Channels | RL | Post-training | Sparse | N/A | N/A | Hard Pruning |
| GraSP | $W$ | $W \circ (H\nabla_W)$ | Before Training | Sparse | N/A | N/A | Hard Pruning |
| LayerDrop | Layers | Random | In-training | Subnetwork | N/A | N/A | Stochastic Hard Pruning |
| SDGP | $\nabla_W$ | $|\nabla_W|$ | In-training | Dense | N/A | N/A | M:N Hard Pruning |
| DropBack | $W$ | $|\nabla_W|$ | In-training | Dense | Reset to $W_0$ | N/A | No Decay |
| Procrustes | $W$ | $|\nabla_W|$ | In-training | Sparse | Reset to $W_0$ | N/A | Exp. Decay $\lambda^t$ |
| LTH | $W$ | $|W|$ | Retraining | Sparse | Rewind to $W_k$ | LMC (Offline) | Hard Pruning |
| **GDSTAR** (ours) | $W$ | $|\nabla_W|$ | In-training | Sparse | Rewind to $W_k$ | Frobenius (Online) | Exp. Decay $\lambda^t$ |

2018). Instead of relying on importance scores computed post-training, GraSP (Wang et al., 2020) estimates the impact of weight removal using gradient information before training begins. It evaluates weight importance only at initialization, limiting its effectiveness for deeper networks where weight significance evolves during training.

A growing body of research has therefore shifted toward pruning during training (McDanel et al., 2022; Golub et al., 2020; Yang et al., 2020; Molchanov et al., 2017a; Fan et al., 2020), which we refer to as sparse training. Some sparse training methods primarily act as regularizers to improve generalization rather than producing a pruned model for inference (Louizos et al., 2018; Molchanov et al., 2017a). For example, LayerDrop (Fan et al., 2020) randomly removes layers during training. SDGP (McDanel et al., 2022) applies structured pruning based on gradient magnitudes $|\frac{dL}{dO}|$, zeroing out the least significant gradients in M:N patterns (i.e., preserving M out of every N consecutive gradients) to improve hardware efficiency (NVIDIA Corporation, 2020). DropBack (Golub et al., 2020) focuses on the magnitudes of accumulated gradients rather than individual gradient values. Instead of permanently zeroing out gradients, it resets weights with small accumulated gradients to their initial values $W_0$, enabling training under a constrained weight budget while reducing multiply-accumulate (MAC) operations (Golub et al., 2020).

In contrast, other sparse training approaches explicitly remove weights during training to produce pruned learning models also suitable for sparse inference. Procrustes (Yang et al., 2020) extends DropBack (Golub et al., 2020) by combining weight resetting with a decay process. As shown in Algorithm 2 (see Appendix), it resets weights with small accumulated gradients to their initial values ($W_0$) while applying an exponential decay factor $\lambda^t$, enforcing sparsity over training. Previous sparse training methods like DropBack and Procrustes rely on resetting selected weights to their initial values ($W_0$) to stabilize training (Golub et al., 2020; Yang et al., 2020). According to LTH (Frankle et al., 2020a) while this strategy is effective for small networks, it becomes suboptimal for larger and deeper models because early SGD steps introduce noise and instability (Frankle et al., 2020a). LTH proposes identifying sparse subnetworks by rewinding weights to intermediate checkpoints ($W_k$) rather than $W_0$ (Frankle et al., 2020a). The effectiveness of rewinding depends heavily on identifying an appropriate epoch $k$. As shown in Algorithm 3 (see Appendix), Linear Mode Connectivity (LMC) (Frankle et al., 2020a) provides an offline procedure to evaluate candidate epochs by retraining clones of the model on different data orders and interpolating between them. Considering the consistency in validation loss, the epoch that minimizes the mean loss across interpolated models is chosen as the rewind point. LTH highlight the promise of rewinding but remain limited by offline analysis and costly prune-retrain cycles (Renda et al., 2020).

Beyond the above pruning-based and reset/rewind methods, recent dynamic sparse training (DST) approaches introduce mechanisms such as gradient-based regrowth, plasticity-aware connectivity updates, or alternating compression cycles (Zhu & Gupta, 2017; Mocanu et al., 2018; Evci et al., 2020; Jayakumar et al., 2020; Liu et al., 2021; Yin et al., 2023; Ji et al., 2024b; Li et al., 2024; Ji et al., 2024a). These works target full DST pipelines with active regrowth and connectivity evolution, whereas our focus is to identify effective rewind points online during a single training run, extending the rewinding mechanism from LTH without performing an exhaustive prune-retrain cycle.

To evaluate the impact of reset ($W_0$) and rewind ($W_k$) strategies, we analyze the Top-1 test accuracy of different DNNs in Figure 1. $Sp.$ and $k$ indicate the achieved sparsity and the optimal rewind point (iteration for the first three models and epoch for the last two), respectively. Figure 1a compares the achieved accuracy to the dense models. We observe that on LeNet300 (Frankle & Carbin, 2018), both approaches perform comparably to the dense model, whereas on more complex DNNs, reset yields lower accuracy compared to rewind. For deeper analysis of the rewinding effect, we compare the accuracy of two models, LeNet300 (simple model) and ResNet50 (He et al., 2016) (complex model), across different rewind points in Figure 1b and 1c. For LeNet300, accuracy remains stable across different rewind iterations, suggesting that a simple reset strategy suffices. In contrast, for ResNet50, identifying an appropriate $k$ significantly improves performance, highlighting the necessity of adaptive rewinding in stabilized pruning of larger models.

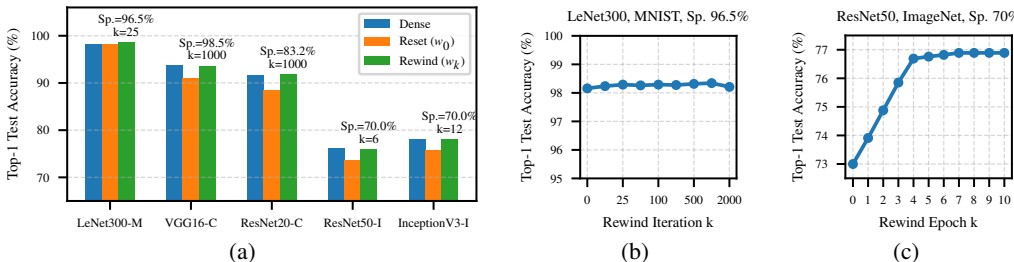

Figure 1: Test accuracy of dense and sparse DNNs under reset and rewind strategies.

Beyond identifying appropriate rewind points, selecting the right weights for removal is critical in sparse training. A common method is to remove weights with smaller absolute values. While this approach is effective in post-training pruning, it is unreliable during training because weights are still adapting and their magnitudes may not reflect eventual importance (McDanel et al., 2022). An alternative criterion is the use of gradient magnitudes (Siems et al., 2021). Gradients encode both the direction and magnitude of weight updates. Weights with consistently small gradients contribute little to learning and can therefore be pruned with minimal impact on accuracy. Figure 2 depicts the distribution of accumulated gradient magnitudes in LeNet300 training on MNIST. We observe that a large fraction of weights exhibit near-zero accumulated gradients. These weights receive no meaningful gradient signal, do not learn useful representations, and can be selected for pruning.

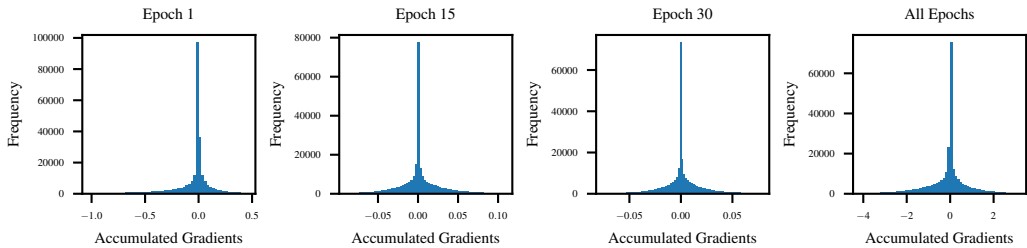

Figure 2: Accumulated gradients distribution in MNIST training with LeNet300.

Existing sparse training methods face several limitations, including degraded accuracy at high sparsity, unreliable pruning criteria during training, difficulty in stabilizing large-scale models, and high computational overhead. This motivates us to propose GDSTAR, a practical and efficient sparse training framework that (1) achieves high sparsity without degrading accuracy or destabilizing optimization, (2) select the weights to be pruned based on reliable training-time criteria, (3) adaptively controls the pruning rate to balance early sparsity with the preservation of essential weights, and (4) integrates pruning into the training loop with minimal overhead. Our approach leverages gradient-based statistics, which both guide weight selection and provide a dynamic signal to identify stable rewind points, thereby improving scalability for large-scale DNNs while avoiding costly offline analysis. Since gradients are continuously available during backpropagation as a part of training, GDSTAR makes adaptive pruning decisions on the fly with minimal computational overhead. Furthermore, to enable early sparsity without sacrificing performance, we employ a pruning scheduler, which determines the number of weights pruned at each epoch, and gradually decay selected weights, ensuring a smooth and stable optimization process.

## 3 GRADIENT-BASED DYNAMIC SPARSE TRAINING WITH ADAPTIVE REWINDING

We propose a Gradient-based Dynamic Sparse Training method with Adaptive Rewinding that effectively supports models of different sizes and complexities without offline retraining. GDSTAR follows three main steps: (1) rewind point identification (§ 3.1), (2) weight selection (§ 3.2), and (3) decay-based weight pruning (§ 3.3). First, it dynamically identifies rewind points by computing the Frobenius norm of gradients at each epoch. The epoch with the minimum value is selected, and its weights are stored and dynamically updated throughout training. Second, it selects weights for pruning based on their accumulated squared gradients within each epoch. Weights with the smallest accumulated squared gradients are selected according to a dynamic pruning rate scheduler. To reduce overhead, gradients are partitioned into chunks, and averaged quantiles are used instead of full sorting. Third, it prunes the selected weights using a dynamic decay process that gradually reduces them toward zero, with a pruning threshold removing those already near zero.

### 3.1 REWIND POINT IDENTIFICATION

Determining the optimal rewind point in a way that integrates directly into the training loop is crucial for effective sparse training. Our approach is based on the idea of measuring network stability (Frankle et al., 2020a) during training using gradient noise. At each epoch, we evaluate a stability metric and dynamically track the point at which the model is most stable (i.e., exhibits the minimum gradient noise), thereby avoiding costly offline analysis while relying only on information already available during training. To capture stability, we aggregate the magnitude of gradients across all weights $W$ within each epoch $k$. Simply summing raw gradients is unreliable, since positive and negative values can cancel out (e.g., gradients of $+0.5$ and $-0.5$). To avoid this, we consider aggregation schemes that eliminate sign cancellation. One option is to use the sum of absolute values, corresponding to the $\ell_1$-norm (Equation 1). $k$ denotes the epoch with $N_k$ mini-batch iterations and $W$ denotes the set of weights. However, this approach is non-smooth at zero, making it less suitable in deep learning contexts (Goodfellow et al., 2016).

$$S_k^{(\text{abs})} = \sum_{i=1}^{N_k} \left\| \nabla_W \mathcal{L}_k^{(i)} \right\|_1 = \sum_{i=1}^{N_k} \sum_{w_j \in W} \left| \nabla_{w_j} \mathcal{L}_k^{(i)} \right| \tag{1}$$

Instead, we adopt the squared-gradient formulation, which for each weight $(w_j)$ accumulates squared gradients across all mini-batches in epoch $k$ (Equations 2) and then sums them across weights (Equations 3). This formulation is differentiable and consistent with the method we employ later for weight selection (§ 3.2), ensuring coherence in our sparse training framework.

$$G_{k,j}^{(\text{sq})} = \sum_{i=1}^{N_k} \left( \nabla_{w_j} \mathcal{L}_k^{(i)} \right)^2 \tag{2}$$

$$S_k^{(\text{sq})} = \sum_{w_j \in W} G_{k,j}^{(\text{sq})} = \sum_{w_j \in W} \sum_{i=1}^{N_k} \left( \nabla_{w_j} \mathcal{L}_k^{(i)} \right)^2 = \left\| \nabla_W \mathcal{L}_k^{(i)} \right\|_F^2 \tag{3}$$

To restore the scale of the original gradients, we take the square root of the squared gradients, yielding the $\ell_2$-norm, or Frobenius norm, of gradients (Equations 4). To show how the Frobenius norm is more precise than the squared gradients, we compare their values across different epochs in Figure 6 (see Appendix § A.3). We observe that the Frobenius norm shows the minimum variations between epochs 200 and 300 more clearly. The squared gradients metric (no square root) grows quadratically, becoming overly sensitive to outliers and making variations in the early or final epochs harder to detect. The full procedure is summarized in Algorithm 4 (see Appendix).

$$S_k^{(\text{F})} = \left\| \nabla_W \mathcal{L}_k \right\|_F = \sqrt{S_k^{(\text{sq})}} \tag{4}$$

This Frobenius norm can be interpreted as the overall amount of learning effort within an epoch: lower values correspond to reduced gradient noise and higher stability. We then maintain a running minimum of this value across epochs, as shown by dot markers in Figure 3. Whenever a new minimum is observed, the corresponding epoch $k$ is marked as the candidate optimal rewind point $k^*$, and its weights $W_{k^*}$ are saved as the optimal rewind weights. This process is performed online during training without any additional runs or offline computations.

The Frobenius norm of gradients is a runtime identifier for the rewind point. To show this, we compute the Frobenius norm of gradients across several models and plot its evolution over training epochs, shown in Figure 3. Dot markers indicate running minima. The optimizer used is SGD with a cosine annealing learning rate scheduler. We make three key observations. First, for simpler models such as LeNet (LeCun et al., 1998a), the minimum Frobenius norm occurs at epoch $k^* = 0$ and then follows an overall incremental trend throughout training. Second, for more complex models such as MobileNetV2 (Sandler et al., 2018), the minimum continues to update until around $k^* = 10$, after which it remains stable until approximately $k^* = 274$. Beyond this point, the running minimum is updated again as training progresses. At these stages, resetting to $W_0$ becomes disruptive, and it is necessary to rewind to weights closer to the current epoch to stabilize training. We observe a similar pattern in VGG16 (Simonyan & Zisserman, 2015), InceptionV3 (Szegedy et al., 2016), ResNet18 (He et al., 2016), and DenseNet121 (Huang et al., 2017). Third, for medium-scale models such as EfficientNetB0 (Tan & Le, 2019) and ShuffleNetV2 (Ma et al., 2018), the minimum Frobenius norm is updated only during the early epochs and, unlike the previously mentioned networks,

remains unchanged in the later stages of training. These results align with LTH findings (Frankle et al., 2020a) and support the effectiveness of using the Frobenius norm of gradients as a proxy for identifying rewind points, while maintaining accuracy (see § 5.1 for more details on accuracy).

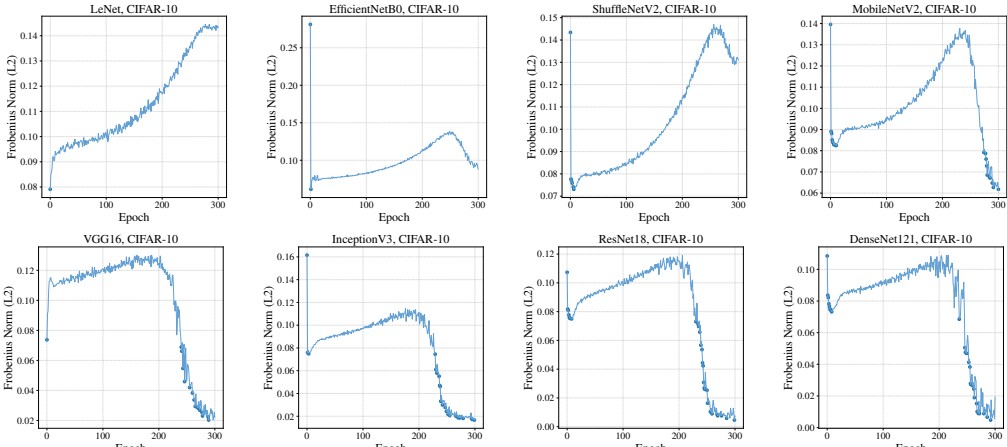

Figure 3: Comparison of the Frobenius norm of gradients across different models.

## 3.2 WEIGHT SELECTION

A critical challenge in sparse training is deciding which weights to prune (McDanel et al., 2022). Our method tackles this by ranking weights based on their accumulated squared gradient magnitudes within each epoch. Intuitively, this measure captures each weight's contribution to learning: weights that consistently receive little to no gradient signal are less important and thus become pruning candidates (McDanel et al., 2022). Most of these values concentrate near zero and can be safely removed (Golub et al., 2020), as discussed in § 2.

Algorithm 5 (see Appendix) summarizes the weight selection procedure. For each weight $w_j$ in epoch $k$ with $N_k$ batches, we compute the squared gradient accumulation $G_{k,j}^{(\text{sq})}$ (Equation 2) across all iterations in the epoch. This criterion is used in both rewind point identification and weight selection, integrating into the training loop with minimal overhead. To decide how many weights to prune, we use a prune rate $\alpha_k \in (0, 1)$, specifying the fraction of weights targeted at epoch $k$. The pruning threshold $\tau_k$ is then set as the $\alpha_k$-quantile of the $G_{k,j}^{(\text{sq})}$ distribution, at the end of epoch $k$, after all iterations are complete. According to Equation 5, all weights whose accumulated squared gradient ($G$ value) is smaller than the threshold $\tau_k$ are considered less important and marked for pruning. The weight pruning procedure is detailed in the next subsection (§ 3.3).

$$W_{\mathcal{T}_k} = \{w_j \mid G_{k,j}^{(\text{sq})} < \tau_k\} \tag{5}$$

Calculating $\tau_k$ requires sorting all elements by their values. Since DNNs contain an enormous number of weights, this sorting step comes with very high memory and computational costs. To address this, we use chunked quantile approximation. We partition the set of weights into $C$ chunks, For each chunk $c = 1, \ldots, C$, we compute local quantiles of $G_{k,j}^{(\text{sq})}$ (Equation 6), and then define the global threshold as the average of these quantiles across all chunks (Equation 7). This approximation allows us to efficiently determine which weights should be targeted (Equation 5). We will discuss the effectiveness of our weight selection criteria and the choice of $C$ and in § A.3 (see Appendix).

$$q_k^{(c)} = \text{Quantile}_{\alpha_k}\left(\{G_{k,j}^{(\text{sq})} : w_j \in \text{chunk } c\}\right) \tag{6}$$

$$\tau_k = \frac{1}{C} \sum_{c=1}^{C} q_k^{(c)} \tag{7}$$

Finally, to regulate pruning dynamics, we schedule $\alpha_k$ (a.k.a., prune rate) across epochs. A constant rate across all epochs may be too aggressive or too weak. If $\alpha$ is set too low, the network fails to achieve meaningful sparsity, while excessively high values risk disrupting the network structure

and significantly degrading accuracy. Thus, $\alpha$ serves as a critical hyperparameter in balancing the sparsity-accuracy trade-off. Its role is analogous to the learning rate in optimization: just as learning rate scheduling influences convergence behavior (Goodfellow et al., 2016), prune rate scheduling governs the dynamics of sparsity throughout training. Consequently, we adopt a linear scheduler, defined in Equation 8, where $k$ is the current epoch, $K$ is the total number of training epochs, $\alpha_0$ is the initial prune rate, and $\alpha_K$ is the final prune rate.

$$\alpha_k = \alpha_0 + \frac{k}{K}\left(\alpha_K - \alpha_0\right) \tag{8}$$

Depending on the choice of $\alpha_0$ and $\alpha_K$, this scheduler can be increasing, decreasing, or constant. Our experimental results (see § 5.1) show that a decreasing linear scheduler gradually relaxes pruning toward the end of training, balancing sparsity with convergence stability. Our linearly decreasing scheduler is designed to (1) increase sparsity early while retaining the model's scaffolding, (2) allow the model to remain sparse for the majority of epochs, thus reducing computation, and (3) apply pruning conservatively in the final epochs to preserve convergence and accuracy.

Although GDSTAR is designed for unstructured sparsity, its selection criterion naturally extends to structured sparsity. By aggregating per-weight stability statistics into a group-level score (e.g., max/mean/norm of the per-weight statistics), entire blocks, channels, or n:m patterns can be pruned when the aggregated signal falls below a threshold, enabling hardware-friendly structured sparsity.

### 3.3 DECAY-BASED WEIGHT PRUNING

Once the optimal rewind weights ($W_{k^*}$) and target weights ($W_{\mathcal{T}_k}$) are identified, we perform pruning through a decay-based update rule. Instead of directly assigning the selected weights to their corresponding optimal rewind weights or removing or decaying weights, we adopt a 3-step decay-based weight pruning process as defined in Equation 9. (1) Assigning the selected weights to their rewind values, ensuring stability (Frankle et al., 2020a). (2) Gradually decaying these values using an exponential factor $\lambda^t$, which smoothly reduces weights toward zero over time. (3) Explicitly setting near-zero weights to zero using a small threshold $\epsilon$. Note that we track the history of weights that have been eliminated in a mask and ensure that they remain excluded throughout training, so they do not participate in any computations during forward or backward propagation.

Algorithm 6 (see Appendix) shows the decay-based weight pruning process. $W_{k^*}$ is the optimal rewind weights and $W_{\mathcal{T}_k} \subseteq W$ is the set of weights selected for pruning at epoch $k$. For each weight $w_j \in W_{\mathcal{T}_k}$, $w_j^*$ is its corresponding optimal rewind weight in $W_{k^*}$. The updated value of $w_j$ at the end of epoch $k$ is given in Equation 9, where $\lambda \in (0,1)$ is the decay rate and $t$ is the decay step. The decay rate $t$ controls how quickly weights decay toward zero. This parameter can be defined either as a constant or dynamically linked to the current epoch, iteration (Yang et al., 2020), or rewind epoch. Simply setting $t$ equal to the current epoch makes the rewind weights ineffective. For example, if the rewind point $k^*$ is identified in later epochs, then $t$ becomes large, causing $\lambda^t$ to approach zero and thereby suppressing the contribution of $W_{k^*}$. To address this issue, we define $t$ as the difference between the current epoch and the optimal rewind epoch, i.e., $t = k - k^*$. By this, $t$ resets to zero whenever $k^*$ is updated, ensuring that $W_{k^*}$ initially has a strong influence. As training continues without a new $k^*$, $t$ increases with each epoch, and the influence of the rewind weights gradually diminishes. We discuss the choice of $t$ and $\lambda$ in § 5.1.

$$w_j = \left(w_j^* \lambda^t\right) \cdot \mathbb{1}(|w_j^* \lambda^t| \geq \epsilon). \tag{9}$$

This 3-step decay-based pruning process offers three benefits: (1) exploiting stability from the rewind point (Frankle et al., 2020a), unlike prior approaches that reset to the initial weights (Golub et al., 2020; Yang et al., 2020), (2) smooth transition to sparsity without disrupting convergence that may occur with hard resets or aggressive decay alone, and (3) a regularization effect that improves generalization (Louizos et al., 2018). The use of rewind weights, exponential decay, and a pruning threshold ensures a stable, gradual, and efficient pruning process fully integrated into training.

## 4 METHODOLOGY

We consider eleven DNNs (see Table 2) trained on two datasets: LeNet300 (Frankle & Carbin, 2018) on MNIST (LeCun et al., 1998b) and all other models on CIFAR-10 (Krizhevsky & Hinton, 2009). All models are trained for 300 epochs with a batch size of 128 for training and 100 for testing on NVIDIA P100 GPUs. We use SGD with momentum 0.9 and weight decay $5 \times 10^{-4}$, with

a base learning rate of $0.05$ and standard cross-entropy loss function. We employ a cosine annealing scheduler to decay the learning rate. The pruning algorithm is configured with an initial prune rate of $\alpha_0 = 0.95$, gradually decayed to a final value of $\alpha_K = 0.25$. The decay rate is $\lambda = 0.75$, and the pruning threshold is fixed at $\epsilon = 5.5 \times 10^{-5}$. The number of chunks is set to $C = 1$ for all evaluated models except VGG19BN (Simonyan & Zisserman, 2015), ResNet34, and ResNet50 (He et al., 2016), which require $C = 2$ due to their higher parameter counts.

To evaluate the effectiveness of GDSTAR, we report two metrics: (1) Top-1 test accuracy, measuring predictive performance, and (2) sparsity, measuring the compression effectiveness of pruning. We design the following comparisons. (1) We compare pruned DNNs against their corresponding dense counterparts to assess how well GDSTAR achieves sparsity while preserving accuracy. (2) We compare the GDSTAR rewind strategy ($W_{k^*}$), identified using the Frobenius norm of gradients, against the reset-to-initialization strategy ($W_0$) as proposed in Procrustes (Yang et al., 2020), a state-of-the-art sparse training method. This validates the effectiveness of our proposed rewind-point identification method. We quantitatively analyze the effect of individual design choices (decay rate, prune rate, prune threshold, and chunked quantile) in GDSTAR by varying its hyperparameters. The results of chunked quantile (§ A.3) and the prune threshold (§ A.3) are shown in the Appendix.

## 5 EVALUATION

### 5.1 ACCURACY & SPARSITY ANALYSIS

We evaluate the accuracy–sparsity trade-off achieved by GDSTAR in comparison with dense models and with the reset strategy used in Procrustes (Yang et al., 2020). Table 2 reports the Top-1 test accuracy and the sparsity levels obtained for each DNN. We make three key observations. First, across all of our experiments and under the same sparsity ratios, our rewind strategy consistently outperforms the reset strategy in terms of accuracy. On average, GDSTAR improves accuracy by $0.72\%$ with the maximum gain of $2.13\%$ observed on EfficientNetB0 (Tan & Le, 2019). Second, the accuracy of pruned models under GDSTAR remains comparable to that of dense models. At an average sparsity of $93.64\%$ (up to $96\%$ for VGG19BN, ResNet34, and EfficientNetB0), the average accuracy drop is negligible, only $0.94\%$. The only notable degradation occurs in LeNet ($9.27\%$ at $91\%$ sparsity). Since LeNet (LeCun et al., 1998a) is a very small model with limited capacity, such a high pruning ratio severely reduces its performance, causing a significant accuracy drop. Therefore, we treat it as an outlier and exclude it from the average accuracy drop. Third, in some cases, pruning with GDSTAR even surpasses the dense baseline. For instance, VGG19BN (Simonyan & Zisserman, 2015) at $96\%$ sparsity achieves $0.13\%$ higher accuracy than its dense counterpart, which we attribute to the implicit regularization effect (Louizos et al., 2018) of combining weight decay with rewinding to more stable points (Frankle et al., 2020a) during training.

In summary, GDSTAR achieves high sparsity levels (over $93\%$ on average) while maintaining accuracy comparable to dense models and consistently outperforming Procrustes. These findings confirm the effectiveness of our rewind strategy, where identifying rewind points via the Frobenius norm of gradients provides a more effective trade-off between compression and accuracy in sparse training.

Table 2: Comparison of DNNs performance under dense and sparse settings.

| Model | Sparsity (%) | Accuracy (%) | | |
|---|---|---|---|---|
| | | Dense | Reset to $W_0$ | **Rewind to $W_{k^*}$ (GDSTAR)** |
| LeNet300 | 92 | 98.56 | 97.92 | **98.29** |
| LeNet | 91 | 77.16 | 66.12 | **67.89** |
| VGG16 | 95 | 93.07 | 92.62 | **92.68** |
| VGG19BN | 96 | 93.92 | 93.55 | **94.05** |
| InceptionV3 | 92 | 94.09 | 91.85 | **92.43** |
| ResNet18 | 94 | 95.25 | 94.62 | **94.67** |
| ResNet34 | 96 | 95.28 | 94.57 | **94.92** |
| ShuffleNetV2 | 92 | 90.50 | 86.98 | **88.51** |
| EfficientNetB0 | 96 | 91.92 | 88.08 | **90.21** |
| MobileNetV2 | 93 | 93.68 | 93.00 | **93.51** |
| DenseNetC | 93 | 95.23 | 91.67 | **91.79** |

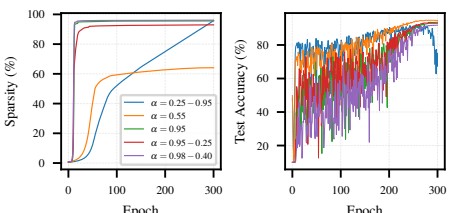 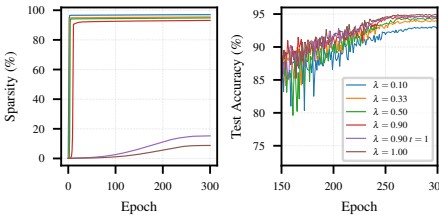

Figure 4: Impact of prune rate $\alpha$ on MobileNetV2.  Figure 5: Impact of decay rate $\lambda$ on ResNet18.

## 5.2 IMPACT OF THE PRUNE RATE

The prune rate ($\alpha$) specifies the fraction of weights removed at each epoch and directly controls the sparsity–accuracy trade-off. Closely related is the decay step ($t$), which controls how aggressively the selected weights are decayed across epochs. Together, $\alpha$ and $t$ influence not only the timing of when sparsity emerges but also the stability of optimization across training (Yang et al., 2020).

Figure 4 shows the effect of $\alpha$ by varying its scheduling strategy across epochs on both achieved sparsity and final model accuracy. Specifically, we compare (i) decreasing $\alpha$ (from 0.95 to 0.25), (ii) increasing $\alpha$ (from 0.25 to 0.95), (iii) high constant $\alpha$, when $\alpha$ is kept constant at a high value (e.g., 0.95), and (iv) low constant $\alpha$ (0.55). We make three key observations. First, decreasing $\alpha$ achieves high sparsity early while preserving accuracy in later epochs. This schedule maintains performance stability, since pruning is aggressive at the start but gradually relaxes toward the end (red line). This indicates that the starting and ending values of $\alpha$ effectively manage the sparsity-accuracy trade-off. Second, increasing $\alpha$ (e.g., from 0.25 to 0.95) delays sparsity, leading to disruptive pruning late in training and a sharp performance drop (blue line). Third, a high constant $\alpha$ (e.g., 0.95) produces early sparsity with little impact on final accuracy, since most weights are removed in the early epochs and later pruning (the pruning process in the final epochs) has minimal effect (green line). This demonstrates that the starting value of $\alpha$ plays a crucial role in determining the pruning strategy, while the final value of $\alpha$ becomes less significant when the initial $\alpha$ is high. Fourth, a low constant $\alpha$ (e.g., 0.55) results in insufficient sparsity (around 64.11%), falling short of compression objectives (yellow line). In summary, the initial $\alpha$ is more important than its final value. High early pruning rate increases sparsity by selecting many weights, while small early $t$ avoids overly aggressive reduction and preserves the model's initial scaffolding. Gradually decreasing $\alpha$ so that fewer weights are pruned in later epochs stabilizes convergence and maintains accuracy.

## 5.3 IMPACT OF THE DECAY RATE

The decay rate ($\lambda$) directly controls how aggressively selected weights are reduced during pruning, making it a key factor in the trade-off between sparsity and accuracy. Its small values accelerate weight decay, leading to rapid sparsity growth but risking accuracy loss and model generalization. Its larger values preserve weights longer, stabilizing training at the expense of slower sparsity gains. Figure 5 evaluates the impact of $\lambda$. We observe five main patterns. First, using aggressive decay ($\lambda = 0.1$), sparsity rapidly reaches 96.5% by epoch 4 and 97.02% by the end of training. However, this incurs a 1.6% accuracy drop compared to $\lambda = 0.9$, showing that early over-pruning harms generalization. Second, using gradual decay ($\lambda = 0.9$), sparsity grows more slowly, reaching 90.5% at epoch 12 and converging at 93.13%. Accuracy is best preserved (94.69%), but sparsity remains lower. Third, balanced decay ($\lambda = 0.5$) provides the best compromise, achieving 94.82% sparsity with only a 0.29% accuracy drop relative to $\lambda = 0.9$. Fourth, using $\lambda = 1.0$, which corresponds to no decay, sparsity remains negligible (8.76%), essentially dictated by the pruning threshold rather than the decay mechanism. This shows that decay is essential for achieving meaningful sparsity. Fifth, the exponentiation nature of the decay process, indicated by decay step ($t$), plays a crucial role. Ignoring the exponentiation by simply setting $t = 1$ leads to very limited sparsity. In sum, intermediate decay rates ($\lambda = 0.5$–$0.9$) offer high sparsity while maintaining accuracy.

## 6 CONCLUSION

We propose GDSTAR, a gradient-driven sparse training framework with adaptive rewinding, designed to eliminate costly offline retraining while remaining effective across both small and large models. It determines stable rewind points via the Frobenius norm of gradients, selects weights with the smallest accumulated squared gradient magnitudes, and applies pruning progressively using a decay process guided by a dynamic pruning rate. GDSTAR achieves up to 96% sparsity with an average accuracy drop of only 0.94% relative to dense models. GDSTAR outperforms the accuracy of state-of-the-art sparse training by 0.72% at the same sparsity level.

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

# A APPENDIX

## A.1 BACKGROUND ALGORITHMS

---

**Algorithm 1** Simplified overview of DNN training

---

1: $W^{(0)} \sim \mathcal{N}(0, \sigma)$                          ▷ Initialize weights
2: **for** each mini-batch $(x, y)$ from dataset **do**
3:     $\hat{y} \leftarrow f(x; W)$           ▷ Compute predictions using forward propagation
4:     $\mathcal{L} \leftarrow \ell(\hat{y}, y)$                            ▷ Compute loss
5:     Compute gradients: $\nabla_{\mathbf{W}} \mathcal{L}$                   ▷ Backpropagation
6:     $W \leftarrow W - \eta \nabla_W \mathcal{L}$                   ▷ Update weights
7: **end for**

---

**Algorithm 2** Procrustes (DropBack with weight decay $\lambda^t$)

---

1: $W^{(0)} \sim \mathcal{N}(0, \sigma)$                          ▷ Initialize weights
2: **while** not converged **do**
3:     $T \leftarrow \left\{ \left| \sum_{i=0}^{t-1} \eta \nabla_w f(W^{(i)}; x^{(i)}) \right| : w \in W_{\text{trk}} \right\}$ ▷ Accumulate gradients for tracked weights
4:     $P \leftarrow \left\{ \left| \eta \nabla_w f(W^{(t-1)}; x^{(t-1)}) \right| : w \in W_{\text{prn}} \right\}$     ▷ Latest gradients for pruned weights
5:     $S \leftarrow \text{sort}(T \cup P)$                         ▷ Combine and sort
6:     $m \leftarrow \mathbb{1}(S > S[k])$            ▷ Create mask for top-$k$ gradients to keep
7:     $W^{(t)} \leftarrow m \odot \left( W^{(t-1)} - \eta \nabla_W f(W^{(t-1)}; x^{(t-1)}) \right)$     ▷ Update unpruned weights
          $+ \overline{m} \odot \lambda^t W^{(0)}$                     ▷ Decay selected weights
8:     $t \leftarrow t + 1$
9: **end while**

---

**Algorithm 3** Rewinding epoch selection using LMC

---

**Require:** model $\mathcal{M}$, dataset $\mathcal{D}$, total epochs $E$, candidate epochs $K$, interpolation steps $T$
1: $\mathcal{S} \leftarrow [\,]$                            ▷ Initialize scores list
2: **for** each $k \in K$ **do**
3:     Train $\mathcal{M}$ for $k$ epochs on $\mathcal{D}$ and save weights as $\theta_k$
4:     $\mathcal{M}_1 \leftarrow \theta_k, \mathcal{M}_2 \leftarrow \theta_k$                    ▷ Create model copies
5:     Shuffle data to get $\mathcal{D}_1$ and $\mathcal{D}_2$
6:     Train $\mathcal{M}_1$ on $\mathcal{D}_1$ for $E - k$ epochs $\rightarrow \theta_1$
7:     Train $\mathcal{M}_2$ on $\mathcal{D}_2$ for $E - k$ epochs $\rightarrow \theta_2$
8:     $\mathcal{L}_k \leftarrow [\,]$                          ▷ Initialize losses list
9:     **for** $\alpha = 0$ to $1$ in $T$ steps **do**
10:       $\theta_\alpha \leftarrow (1 - \alpha)\theta_1 + \alpha\theta_2$          ▷ Create interpolated models
11:       $\ell \leftarrow \text{Loss}(\theta_\alpha, \mathcal{D}_{\text{val}})$                   ▷ Compute loss
12:       Append $\ell$ to $\mathcal{L}_k$
13:     **end for**
14:     $s_k \leftarrow \text{mean}(\mathcal{L}_k)$                      ▷ Compute score
15:     Append $(k, s_k)$ to $\mathcal{S}$
16: **end for**
17: **return** $k^* \leftarrow \arg\min_{(k,s_k) \in \mathcal{S}} s_k$           ▷ Return best rewind epoch

---

## A.2 GDSTAR ALGORITHMS

---

**Algorithm 4** Rewind Point Identification

---

1: $m \leftarrow +\infty \quad k^* \leftarrow 0$                $\triangleright$ Initialize running minimum and rewind epoch
2: **for** each epoch $k = 1, 2, \ldots, K$ **do**
3:     $G_{k,j}^{(\mathrm{sq})} \leftarrow 0$                $\triangleright$ Initialize accumulated squared gradients tensor
4:     **for** each batch $i = 1, 2, \ldots, N_k$ **do**
5:        Perform forward and backward pass and obtain gradient tensor $\nabla_W \mathcal{L}_k^{(i)}$
6:        $G_{k,j}^{(\mathrm{sq})} \leftarrow G_{k,j}^{(\mathrm{sq})} + (\nabla_{w_j} \mathcal{L}_k^{(i)})^2$      $\triangleright$ Accumulate squared gradients for each weight
7:     **end for**
8:     $S_k^{(\mathrm{sq})} \leftarrow \sum_{w_j \in W} G_{k,j}^{(\mathrm{sq})}$              $\triangleright$ Sum across weights
9:     $S_k^{(\mathrm{F})} \leftarrow \sqrt{S_k^{(\mathrm{sq})}}$                $\triangleright$ Frobenius norm
10:    **if** $S_k^{(\mathrm{F})} < m$ **then**
11:      $m \leftarrow S_k^{(\mathrm{F})}$                $\triangleright$ Update minimum
12:      $k^* \leftarrow k$                $\triangleright$ Update rewind epoch
13:      $W_{k^*} \leftarrow W$              $\triangleright$ Update rewind weights
14:    **end if**
15: **end for**
16: **return** $k^*, W_{k^*}$           $\triangleright$ Return optimal rewind epoch and weights

---

**Algorithm 5** Weight Selection

---

1: Input: prune rate $\alpha_0$ and $\alpha_K$, number of chunks $C$, number of epochs $K$
2: **for** each epoch $k = 1, 2, \ldots, K$ **do**
3:     $G_{k,j}^{(\mathrm{sq})} \leftarrow 0$              $\triangleright$ Initialize accumulated squared gradients tensor
4:     $\alpha_k \leftarrow \alpha_0 + \frac{k}{K} (\alpha_K - \alpha_0)$        $\triangleright$ Compute prune rate
5:     **for** each batch $i = 1, 2, \ldots, N_k$ **do**
6:        Perform forward and backward pass and obtain gradient tensor $\nabla_W \mathcal{L}_k^{(i)}$
7:        $G_{k,j}^{(\mathrm{sq})} \leftarrow G_{k,j}^{(\mathrm{sq})} + (\nabla_{w_j} \mathcal{L}_k^{(i)})^2$      $\triangleright$ Accumulate squared gradients for each weight
8:     **end for**
9:     Flatten then partition $G_{k,j}^{(\mathrm{sq})}$ into $C$ chunks
10:    **for** each chunk $c = 1, \ldots, C$ **do**
11:      $q_k^{(c)} \leftarrow \mathrm{Quantile}_{\alpha_k}\left(\{G_{k,j}^{(\mathrm{sq})} : w_j \in \text{chunk } c\}\right)$    $\triangleright$ Compute quantile within the chunk
12:    **end for**
13:    $\tau_k \leftarrow \frac{1}{C} \sum_{c=1}^{C} q_k^{(c)}$            $\triangleright$ Compute global threshold
14:    $W_{\mathcal{T}_k} \leftarrow \{w_j \mid G_{k,j}^{(\mathrm{sq})} < \tau_k\}$        $\triangleright$ Target weights for pruning
15: **end for**

---

**Algorithm 6** Exponential Decay Pruning of Selected Weights

---

1: Input: decay rate $\lambda$, prune threshold $\epsilon$, selected weights $\mathcal{T}_k$, rewind weights $W_{k^*}$, epoch $k$
2: $t = k - k^*$              $\triangleright$ Compute decay step $t$
3: **for** each weight $w_j \in W_{\mathcal{T}_k}$ and $w_j^* \in W_{k^*}$ **do**
4:     $w_j \leftarrow w_j^* \cdot \lambda^t$          $\triangleright$ Update the weight using rewind-decay
5:     **if** $|w_j| < \epsilon$ **then**
6:      $w_j \leftarrow 0$             $\triangleright$ Hard prune the weight
7:     **end if**
8: **end for**

---

## A.3 FROBENIUS NORM VS. SQUARED GRADIENTS

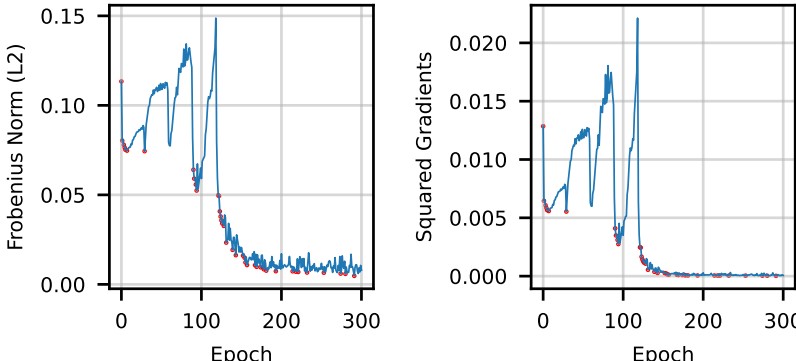

Figure 6: Frobenius norm and squared gradients. Red markers indicate running minima.

## A.4 IMPACT OF THE CHUNKED QUANTILE APPROXIMATION

The chunk parameter ($C$) determines how the weight set is partitioned when estimating quantiles for pruning. Its choice directly affects both efficiency (computation and memory usage) and accuracy (due to quantile approximation error). For a network with $N$ weights, the computational cost of the chunked quantile approximation scales as $O(N \log(N/C))$, while peak memory reduces to $O(N/C)$. Relative savings compared to $C = 1$ are given in Equations 10 and 11.

$$\text{Computation} = \left(1 - \frac{\log(N/C)}{\log N}\right) \times 100\% \tag{10}$$

$$\text{Memory} = (1 - 1/C) \times 100\% \tag{11}$$

Figure 7 evaluates the impact of $C$ by varying its value across a wide range and measure both the training efficiency and final test accuracy.

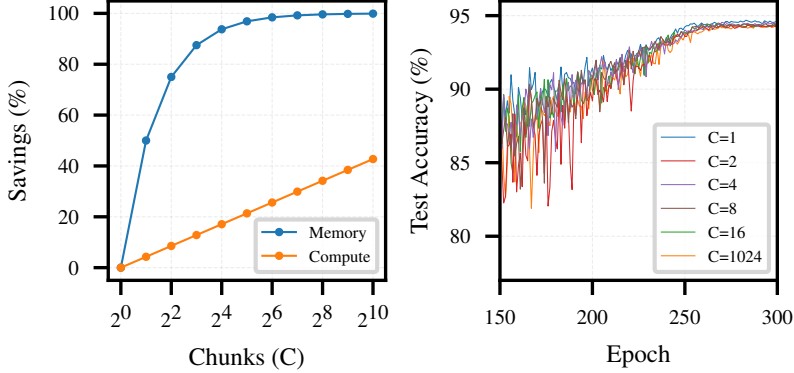

Figure 7: Impact of chunk parameter $C$ on ResNet18.

Our results show three main observations: First, Larger $C$ significantly improves efficiency (reduces computation and memory). For example, $C = 16$ achieves over 93% memory savings and 17% computational savings in quantile calculation (ResNet18 on CIFAR-10 with $N = 11,173,114$ parameters (He et al., 2016)). Chunking is also essential for very large networks (e.g., ResNet34, VGG19BN), where $C = 1$ leads to memory errors, but moderate chunking allows pruning to run

without issues. Increasing $C$ to 2, 4, or higher not only provides the savings, but also resolves this critical restriction. Second, since chunking averages local quantiles rather than computing the true global quantile, the estimation error grows with $C$, slightly degrading accuracy. With $C = 1$, accuracy reaches 94.69%. Accuracy remains stable under small chunk sizes ($C = 4$: 94.56%; $C = 8$: 94.50%; $C = 16$: 94.45%), but drops more noticeably with very large chunking ($C = 1024$: 94.35%). Third, the systematic accuracy drop with larger $C$ highlights the effectiveness of our pruning criterion. If the criterion were weak or random, chunking would not produce consistent accuracy differences.

Chunking provides substantial computational and memory benefits while incurring negligible accuracy loss for moderate values of $C$. Small-to-moderate chunk sizes (e.g., $C = 16$) strike the best balance, enabling scalable pruning in large networks without significantly affecting performance.

## A.5 IMPACT OF THE PRUNE THRESHOLD

The prune threshold ($\epsilon$) determines when decayed weights are permanently set to zero, directly influencing the final sparsity-accuracy balance. Figure 8 shows its impact, revealing three observations. First, using very small thresholds ($\epsilon = 10^{-6}, 10^{-7}$), weights decay slowly, and high sparsity is reached only in later epochs. Accuracy remains high, but computational savings during training are limited. Second, using a large threshold ($\epsilon = 10^{-3}$), weights are pruned too aggressively, reaching 98.72% sparsity, but accuracy drops sharply to 92.91%. Third, using moderate thresholds ($\epsilon = 10^{-4}, 10^{-5}$), provide the best trade-offs. With $\epsilon = 10^{-4}$, sparsity reaches 96.11% at 94.44% accuracy. With $\epsilon = 10^{-5}$, sparsity is slightly lower (93.13%) but accuracy improves to 94.69%. Overall, moderate thresholds strike the best balance, whereas overly large thresholds destabilize learning and overly small thresholds delay sparsity benefits.

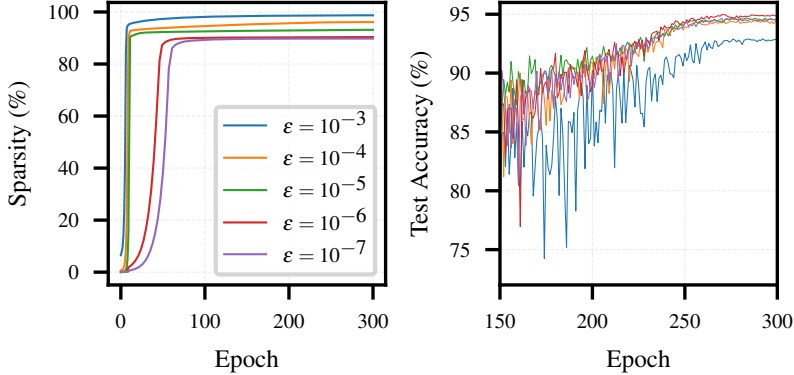

Figure 8: Impact of prune threshold $\epsilon$ on ResNet18.

## A.6 GENERALITY ACROSS MODELS AND DATASETS

To evaluate the generality and robustness of GDSTAR, we consider three complementary model categories that together span a wide range of architectural scales, sparsity regimes, and computational characteristics.

(1) Overoptimized models on CIFAR-10: We first evaluate GDSTAR on overparameterized models for CIFAR-10 (Krizhevsky & Hinton, 2009) (Table 2), including LeNet (LeCun et al., 1998a), VGG (Simonyan & Zisserman, 2015), and ResNet He et al. (2016). These models are intentionally overparameterized for CIFAR-10, a setup that is widely adopted in pruning and rewinding research. This setting provides a controlled environment for isolating the dynamics of adaptive rewinding, without confounding effects from dataset-specific tuning or regrowth heuristics. Under these conditions, GDSTAR enables stable sparse training and consistently outperforms reset-based pruning, supporting the effectiveness of online stability-based rewinding.

(2) Efficiency-oriented models on CIFAR-10: To test whether GDSTAR generalizes beyond overparameterized models, we additionally evaluate lightweight and mobile-class models such as Mo-

bileNetV2 (Sandler et al., 2018), ShuffleNetV2 (Ma et al., 2018), and EfficientNetB0 (Tan & Le, 2019) (see Table 2). These models are not overparameterized for CIFAR-10 and are known to be more sensitive to pruning. We observe that GDSTAR maintains competitive accuracy while achieving high sparsity, demonstrating robustness across structurally diverse model families and validating that the method is not limited to heavy networks.

(3) Larger-scale datasets and models: GDSTAR extends naturally to larger datasets and models because (a) it requires only the gradient tensors already produced during backpropagation, (b) it introduces no architectural modifications to the forward pass, and (c) its pruning dynamics do not depend on model size; the Frobenius-norm stability signal is inherently scale-agnostic.

Due to computational constraints, full ImageNet experiments were not feasible. However, we are actively extending the implementation to ImageNet and larger architectures. As an intermediate step toward large-scale evaluation, we examine ResNet-50 trained on Tiny ImageNet. We train the model with a batch size of 128 for 75 epochs, using quantile chunks $C = 2$ and the default pruning hyperparameters. As shown in Table 3, GDSTAR achieves competitive top-1 accuracy at sparsity levels of 75%, 80%, and 85%, with performance remaining close to the dense baseline. These results demonstrate scalability of GDSTAR.

Table 3: Top-1 test accuracy of GDSTAR on Tiny ImageNet with ResNet50 at different sparsity levels.

| Sparsity | Dense | 75% | 80% | 85% |
|---|---|---|---|---|
| **Top-1 Acc. (%)** | 62.69 | 62.73 | 62.53 | 62.25 |

## A.7 COMPARISON WITH RANDOM REWINDING

To evaluate the effectiveness of the GDSTAR rewinding strategy, which selects rewind points based on the running minimum of the Frobenius norm of gradients, we conduct an ablation study comparing it against a random rewinding baseline. In the random strategy, the rewind point is updated at each epoch with a probability of 10%, selecting that epoch as the new rewind point. We evaluate both approaches on four architectures: LeNet300 (on MNIST) and VGG16, ResNet18, and MobileNetV2 (on CIFAR-10). All comparisons are performed at the same sparsity ratios, as reported in Table 4. For clarity, we note that this ablation study is run for fewer training epochs than Table 2, as its goal is to isolate and compare the rewinding mechanisms rather than to reach full convergence. This difference in training duration may lead to slightly lower final accuracies.

Table 4: Test accuracy improvement of GDSTAR over Random Rewind strategy.

| Model | Sparsity (%) | Test Accuracy (%) | |
|---|---|---|---|
| | | Random Rewind | GDSTAR |
| LeNet300 | 90 | 97.79 | 98.26 |
| VGG16 | 94 | 88.19 | 89.54 |
| ResNet18 | 93.7 | 90.03 | 91.62 |
| MobileNetV2 | 92.8 | 89.96 | 91.66 |

Across all models, GDSTAR achieves higher test accuracy than the random rewinding baseline, with improvements of 0.47%, 1.35%, 1.59%, and 1.70% for the respective models. These gains demonstrate that selecting rewind points based on the Frobenius-norm stability signal provides a more reliable criterion than stochastic selection. Furthermore, the superior performance of GDSTAR compared to resetting to the initial weights ($w_0$), as discussed in § 5.1, reinforces the effectiveness of the stability-driven rewinding mechanism.

## A.8 COMPARISON WITH THE LTH

We compare GDSTAR with the LTH (Frankle et al., 2020a) across three standard benchmarks.

**LeNet300 on MNIST.** GDSTAR achieves a sparsity of 92% with a test accuracy of 98.29%. LTH attains a slightly higher accuracy of 98.34% at a higher sparsity level of 96.5% when rewinding to iteration 1000 (epoch 1).

**ResNet20 on CIFAR-10.** At a matched sparsity of 83.2%, GDSTAR yields 88.31% accuracy, while LTH achieves 91.77% when rewinding to iteration 1000 (epoch 3).

**VGG16 on CIFAR-10.** GDSTAR obtains 92.68% accuracy at 95% sparsity, whereas LTH reaches 93.75% accuracy with 98.5% sparsity when rewinding to iteration 10000 (epoch 26).

Across all evaluated settings, GDSTAR achieves accuracy that is consistently close to the subnetworks identified by the optimal LTH rewinding points. This suggests that the stability signal derived from the Frobenius norm offers a practical and effective proxy for the rewind points discovered via exhaustive LTH search.

Table 5: Comparison of GDSTAR and LTH across three setups.

| Model | Sparsity (%) | | Test Accuracy (%) | |
|---|---|---|---|---|
| | LTH | GDSTAR | LTH | GDSTAR |
| LeNet300 | 96.5 | 92 | 98.34 | 98.29 |
| ResNet20 | 83.2 | 83.2 | 91.77 | 88.31 |
| VGG16 | 98.5 | 95 | 93.75 | 92.68 |

Across all three settings summarized in Table 5, LTH achieves slightly higher test accuracy than GDSTAR. However, LTH functions as a post-training procedure. It requires first training a dense model, then identifying a winning ticket through an exhaustive sweep over candidate rewind points, and finally retraining the selected subnetwork from the chosen initialization. This multi-stage process is computationally expensive. In contrast, GDSTAR integrates pruning directly into the training process, eliminating the need for dense pretraining and avoiding the repeated retraining cycles required by LTH. Moreover, GDSTAR enables sparse training from early epochs, providing efficiency benefits that LTH cannot offer, since LTH necessarily requires a fully dense training phase before sparsification begins. Overall, these results show that GDSTAR achieves accuracy close to the LTH-optimal subnetworks while requiring only a single training run.

## A.9 OVERHEADS

We measure the total training time, memory usage, and computational cost of several models under two settings: (1) standard dense training and (2) training with GDSTAR enabled. This comparison quantifies the additional cost introduced by our method. All experiments were conducted on an NVIDIA P100 GPU, which does not natively accelerate sparse computation. Consequently, none of the measurements include any performance gains that may arise from sparsity itself. This choice is intentional and allows us to isolate the intrinsic overhead of GDSTAR without confounding improvements coming from hardware or kernel-level sparsity support. It is important to note that the potential gains from sparsity are highly dependent on the underlying hardware and software stack. For example, NVIDIA A100 (NVIDIA Corporation, 2020) provide dedicated acceleration for the 2:4 structured sparsity patterns and specialized accelerators, e.g., Eyeriss (Chen et al., 2019), can exploit multiple forms of sparsity to reduce memory footprint, computational load, and end-to-end training time. Because such hardware-specific speedups vary widely, we disable sparsity exploitation to provide a clean and controlled assessment of GDSTAR's overhead.

### A.9.1 TIME OVERHEAD

As shown in Table 6, the additional training time introduced by GDSTAR is small. On average, we observe a 2.53% increase over baseline dense training, with the maximum overhead reaching 8.19%. This modest cost is expected, as GDSTAR reuses gradients already computed during backpropagation and introduces only a lightweight update procedure for selecting the rewind point.

Overall, these results show that the adaptive rewinding mechanism imposes very limited runtime overhead relative to standard training, while still allowing the method to reach high sparsity (often around 90%) early in the training process.

Table 6: Training time comparison between dense baselines and GDSTAR.

| Model | Time (min) | | Overhead (%) |
|---|---|---|---|
| | Baseline | GDSTAR | |
| LeNet300 | 33.871 | 34.326 | 1.34 |
| LeNet | 47.641 | 51.394 | 7.87 |
| VGG16 | 75.793 | 76.142 | 0.46 |
| VGG19BN | 96.393 | 98.446 | 2.13 |
| InceptionV3 | 203.212 | 204.957 | 0.86 |
| ResNet18 | 136.084 | 138.327 | 1.65 |
| ResNet34 | 228.024 | 229.399 | 0.60 |
| ShuffleNetV2 | 98.321 | 106.376 | 8.19 |
| EfficientNetB0 | 170.823 | 173.244 | 1.42 |
| MobileNetV2 | 193.297 | 194.880 | 0.82 |

### A.9.2 MEMORY OVERHEAD

As shown in Table 7, the peak memory overhead introduced by GDSTAR averages 28.07% on CPU and 21.22% on GPU. This overhead primarily stems from storing a single additional rewind checkpoint during the early training phase. After sparsification begins, the active model quickly becomes significantly smaller, and the effective memory footprint decreases accordingly. As a case study, the largest GPU overhead occurs for VGG16 (95.73%), which is expected due to its high parameter count and convolution-heavy architecture. Importantly, in this setting, GDSTAR achieves 95% sparsity during the early training phase while maintaining 92.68% accuracy. Once this high sparsity is reached, the memory usage drops substantially relative to the dense model.

These results indicate that the temporary overhead required to store the rewind checkpoint is modest relative to the large reductions in model size and computational load achieved after pruning begins.

Table 7: Peak memory comparison between dense baselines and GDSTAR.

| Model | Peak Memory (MB) | | | | Overhead (%) | |
|---|---|---|---|---|---|---|
| | CPU | | GPU | | CPU | GPU |
| | Baseline | GDSTAR | Baseline | GDSTAR | | |
| LeNet300 | 1337.61 | 1656.91 | 21.33 | 30.83 | 23.87 | 44.54 |
| LeNet | 1490.05 | 1806.97 | 31.17 | 31.71 | 21.27 | 1.73 |
| VGG16 | 1846.46 | 2283.54 | 436.77 | 854.89 | 23.67 | 95.73 |
| VGG19BN | 2136.75 | 2746.07 | 612.92 | 830.77 | 28.52 | 35.54 |
| InceptionV3 | 2756.55 | 3559.67 | 1127.04 | 1145.92 | 29.13 | 1.68 |
| ResNet18 | 2004.15 | 2561.80 | 749.25 | 847.17 | 27.82 | 13.07 |
| ResNet34 | 2576.51 | 3385.40 | 1227.81 | 1413.88 | 31.39 | 15.15 |
| ShuffleNetV2 | 2505.87 | 3181.70 | 352.96 | 356.12 | 26.97 | 0.90 |
| EfficientNetB0 | 2407.96 | 3141.59 | 1176.84 | 1207.67 | 30.47 | 2.62 |
| MobileNetV2 | 2781.99 | 3828.18 | 1719.15 | 1741.03 | 37.61 | 1.27 |

### A.9.3 COMPUTATIONAL OVERHEAD

Table 8 reports the total number of CUDA kernel calls for each model under dense baseline training and with GDSTAR enabled. As shown, GDSTAR increases the number of kernel launches due

to the lightweight operations used for computing the Frobenius-norm stability signal and updating pruning masks. The overhead ranges from 42.16% (ShuffleNetV2) to 57.01% (MobileNetV2), with an average of approximately 49.04%.

It is important to emphasize that although the number of kernel calls increases, these calls correspond to inexpensive reduction and masking operations that reuse gradients already computed during back-propagation. As a result, the extra kernel launches contribute only marginally to the total runtime (see Table 6). The computational overhead therefore stems from lightweight bookkeeping (mask and statistics updates) rather than expensive numerical operations such as convolutions or matrix multiplications.

Table 8: Total CUDA kernel calls comparison between dense baselines and GDSTAR.

| Model | Total CUDA Kernel Calls ($10^6$) | | Overhead (%) |
|---|---|---|---|
| | Baseline | GDSTAR | |
| LeNet300 | 39.67 | 58.80 | 48.28 |
| LeNet | 56.37 | 82.61 | 46.55 |
| VGG16 | 187.11 | 273.90 | 46.44 |
| VGG19BN | 332.02 | 519.01 | 56.35 |
| InceptionV3 | 656.29 | 958.14 | 46.03 |
| ResNet18 | 350.55 | 515.89 | 47.16 |
| ResNet34 | 617.24 | 911.82 | 47.64 |
| ShuffleNetV2 | 1059.18 | 1505.50 | 42.16 |
| EfficientNetB0 | 1050.63 | 1605.28 | 52.86 |
| MobileNetV2 | 797.42 | 1252.53 | 57.01 |

