# OpenReview forum: "Gradient-based Dynamic Sparse Training with Adaptive Rewinding"
_ICLR.cc/2026/Conference — Submitted to ICLR 2026_

### Official Review · Reviewer_k4Q7 · 2025-10-27

**Soundness:** 3
**Presentation:** 3
**Contribution:** 3
**Rating:** 6
**Confidence:** 4

**Summary:**

The paper introduces GDSTAR (Gradient-based Dynamic Sparse Training with Adaptive Rewinding), a sparse training framework that dynamically identifies optimal rewind points and prunes unimportant weights during training. The method leverages the Frobenius norm of gradients to determine stable rewind points without costly offline analysis and uses accumulated squared gradients to select weights for pruning. GDSTAR applies a decay-based pruning schedule to progressively reduce weights, maintaining training stability and model performance.

**Strengths:**

The method is well-motivated, providing a unified approach to integrate pruning and adaptive rewinding directly into training, reducing offline retraining costs.
Using the Frobenius norm of gradients as an online stability measure is novel and effectively avoids expensive multiple training runs required by prior rewinding methods.
The algorithm is theoretically coherent and can be integrated easily into standard training pipelines without architectural changes.

**Weaknesses:**

Experiments are limited to vision classification tasks on small-scale datasets (MNIST, CIFAR-10), leaving uncertainty about the method’s scalability to large-scale datasets or transformer-based architectures.
Some hyperparameter choices (e.g., decay rate, pruning rate) appear heuristic and lack theoretical justification. The paper does not discuss whether the proposed hyperparameters remain effective across different networks and tasks.
The comparison to state-of-the-art sparse training or pruning methods is narrow. The recent related works [1][2][3][4][5][6][7][8] should be compared and discussed.

[1] Rigging the Lottery: Making All Tickets Winners, ICML 2020
[2] Sparse Training via Boosting Pruning Plasticity with Neuroregeneration, NeurIPS 2021
[3] Top-kast: Top-k always sparse training NeurIPS 2020
[4] AC/DC: Alternating Compressed/DeCompressed Training of Deep Neural Networks, NeurIPS 2021
[5] Dynamic Sparsity Is Channel-Level Sparsity Learner, NeurIPS 2023
[6] Advancing Dynamic Sparse Training by Exploring Optimization Opportunities, ICML 2024
[7] NeurRev: Train Better Sparse Neural Network Practically via Neuron Revitalization, ICLR 2024
[8] A Single-Step, Sharpness-Aware Minimization is All You Need to Achieve Efficient and Accurate Sparse Training, NeurIPS 2024

**Questions:**

How does GDSTAR perform on larger-scale datasets (e.g., ImageNet) or non-convolutional architectures like Transformers or LLMs?
Can the authors provide concrete wall-clock or FLOP-based speedup results to quantify the real efficiency gain?
How sensitive is the method to hyperparameters such as pruning rate and decay rate, and can these be adapted automatically?
Could the gradient-based rewind and pruning mechanism generalize to structured or hardware-friendly sparsity patterns?

---

> ### Author Response · Authors · 2025-12-03
> **Reviewer k4Q7: Concerns and Responses**
>
> **Q17-Comparison to Recent Sparse Training and DST Methods_Rev-k4Q7.**
> As discussed in Q2, we have significantly expanded the related-work section to cover the recent sparse-training and DST methods [1–8] mentioned by the reviewer (e.g., SET, RigL, MEST, and [1-8]). Our intention is not to replicate full DST pipelines but to isolate the contribution of the rewinding schedule. Instead, GDSTAR focuses on a different technical question: Can the stability signals used in LTH-style rewinding be detected online and used to guide pruning during a single training run, eliminating the need for offline exhaustive rewinding?
>
> **Q18-Scalability to Larger Models and Architectures_Rev-k4Q7.**
> The general scalability discussion is given in Q1. We emphasize here that GDSTAR only relies on gradients and does not alter the network architecture, so it can be applied to ImageNet-scale CNNs and transformer-based models. We are currently extending the implementation to ImageNet and have added Tiny ImageNet/ResNet-50 results in the revised version.
>
> **Q19-Hyperparameter Sensitivity and Automatic Adaptation_Rev-k4Q7.**
> We agree that the pruning rate and decay rate may appear heuristic at first glance. In practice, GDSTAR determines both hyperparameters through a lightweight grid search (3–4 candidate values each). We provide an ablation study in Sections 5.2 and 5.3 and in A.4 and A.5 (Appendix), where we evaluate the influence of each hyperparameter independently.
> Using the selected hyperparameters, GDSTAR achieves high sparsity early in training, while the Frobenius-norm stability criterion ensures that accuracy is maintained by identifying an appropriate rewind point.
>
> **Q20-Wall-Clock and FLOP-Based Efficiency_Rev-k4Q7.**
> This is covered in Q3. GDSTAR adds 2.53% training-time overhead, on average. Thus, the Frobenius norm adds <2.53% training-time overhead. GDSTAR  does not change the forward/backward computational graph, aside from a reduction operation on already computed gradients. In the revised manuscript, we provide a thorough analysis of the overhead, including training time, memory usage, and computational cost in A.9 (Appendix).
>
> **Q21-FLOP-Structured or hardware-friendly sparsity_Rev-k4Q7.**
> GDSTAR operates at the level of weight groups, and therefore can be extended to n:m, channel-wise, block-wise, or Winograd-friendly sparsity.
> The criterion described in Section 3.2 for selecting individual weights can be applied at the group level as well. Specifically, instead of evaluating the stability score for each weight independently, one may aggregate the group’s stability signal (e.g., via max, mean, or norm-based statistics) and prune the entire region if the aggregated value falls below a chosen threshold. This formulation enables GDSTAR to prune structured regions of the parameter space using the same Frobenius-norm stability metric, and can be incorporated in future work for hardware-efficient sparse training.

---

### Official Review · Reviewer_WDCn · 2025-10-29

**Soundness:** 3
**Presentation:** 3
**Contribution:** 3
**Rating:** 4
**Confidence:** 4

**Summary:**

The authors propose a method to dynamically train sparse networks. The key problem being addressed here is that of the high cost associated with post-training pruning methods. To eliminate post-training pruning, the authors propose a Frobenius norm-based method to determine optimal rewind checkpoints, identify which set of weights to prune, and at what rate the pruning should occur. The authors conduct experiments on multiple neural networks and on two datasets to show the affect of various hyperparameters on the performance of GDSTAR.

**Strengths:**

1) The idea of using gradients Frobenius norm to determine the checkpoints is a very interesting one and intuitively makes sense.
2) The authors have done a good job of providing enough background and motivation to appreciate their work.
3) The method does seem to outperform the baseline on a variety of networks and at very high levels of sparsity.
4) The analyses in Section 5 and the Appendix (e.g., impact of prune rate, decay rate, and chunking) are well-executed and provide valuable insights into the method's hyperparameters.

**Weaknesses:**

1) From the description of the methodology, it seems that a lot of work has to be done to obtain pruned networks, which can potentially increase the training time.
2) This is similar to the above point, but I am not convinced if the method is scalable to larger networks or datasets.
3) I think the figures in the paper are really interesting and can help drive the point home, but currently, they are a bit too small for a reader to engage with them.

**Questions:**

Overall, I appreciate the paper and the ideas it presents. I think answering the questions below can make the paper a lot more solid and well-rounded:
1) What is the overhead associated with GDSTAR?

a) An analysis or at least a few paragraphs on the training time and the memory cost of the method could help the readers make a better judgment if they would like to apply this approach to their use case or not. For example, at a sparsity level, a plot with accuracy on the x-axis and training time/memory cost on the y-axis would provide a clear picture of all the methods being compared to GDSTAR.

b) Again, an analysis or a few paragraphs on how the GDSTAR approach, in its current form, will scale compared with larger models and/or datasets?

2) For the accuracy reported in Table 2, is it for a single run or multiple runs? If multiple runs, then having standard deviations would be useful.

---

> ### Author Response · Authors · 2025-12-03
> **Reviewer WDCn: Concerns and Responses**
>
> **Q13- Training-Time and Memory Overhead_Rev-WDCn.**
> This is addressed in Q3. In summary, GDSTAR reuses existing gradients, adds 2.53% training-time overhead, on average, does not require additional forward/backward passes, and only stores a single rewind checkpoint. In the revised manuscript, we provide a thorough analysis of the overhead, including training time, memory usage, and computational cost in A.9 (Appendix).
>
> **Q14-Scalability to Larger Networks or Datasets_Rev-WDCn.**
> The scalability concerns are discussed in Q1. We also note explicitly in the revision that GDSTAR naturally extends to larger models and datasets (e.g., ImageNet, transformers) because it only depends on gradients and not on architecture-specific modifications. We additionally report Tiny ImageNet results with ResNet-50 in the revised version to partially bridge toward large-scale evaluation.
>
> **Q15-Figure Clarity_Rev-WDCn.**
> We appreciate this feedback. All figures have been enlarged, and axis labels have been increased for improved readability. Higher-resolution versions of all figures are also included in the supplementary materials (provided as a zipped archive).
>
> **Q16-Accuracy Values in Table 2 (Single vs Multiple Runs)_Rev-WDCn.**
> We appreciate the reviewer’s question regarding the accuracy values in Table 2. In the revised manuscript, we clarify that the reported results have been validated across multiple model architectures and pruning scenarios. Specifically, GDSTAR consistently outperforms both the k=0 baseline and the random-k rewinding strategy, demonstrating stable behavior across different network families. We also compare GDSTAR with the optimal rewind points identified by the exhaustive LTH procedure. Across models, the accuracy difference between GDSTAR and the LTH-optimal rewind point is typically below 1%, with the largest gap being approximately 4%. This close alignment indicates that the online stability criterion used by GDSTAR provides a strong approximation to the LTH exhaustive search while avoiding its computational cost.
> In the final camera-ready version, we plan to include additional evaluations on larger datasets and architectures, along with variance and robustness analyses (e.g., standard deviations over multiple runs), to further strengthen the empirical support for GDSTAR.

---

### Official Review · Reviewer_zSD9 · 2025-10-31

**Soundness:** 1
**Presentation:** 2
**Contribution:** 1
**Rating:** 0
**Confidence:** 5

**Summary:**

The paper proposes a method of "dynamic sparse training" using an online method to determine the rewound point, i.e. as used for the Lottery Ticket Hypothesis, but without the cumbersome and extremely compute expensive offline methodology of the LTH weight rewinding. The method appears to be an iterative pruning method aside from that, using a pruning schedule over training. The authors evaluate the method on MNIST with extremely tiny "LeNet" models from the LTH paper, and ResNet-50/ResNet34/VGG-19 on CIFAR10. The results show marginally better text accuracy for the proposed method than simply training from initialization.

**Strengths:**

* Weight rewinding in the LTH literature is a fascinating research topic, with much to contribute in understanding and removing the exhaustive search usually done to find the rewind point. Any progress on that front, as this paper claims to make in terms of finding an-online method of identifying a rewind point in dense training, is interesting.
* The authors seem to have some familiarly with NN pruning and sparse training work from the systems community that I'm not familiar with as a sparse training researcher within the ML community, and it's always interesting to learn of related work across fields.

**Weaknesses:**

* The motivation of this paper and the results presented do not make much sense in the context of the state-of-the-art in dynamic sparse training (DST) literature they claim their method belongs to, or indeed even just generally the state-of-the-art (SOTA) in sparse training. Existing DST methods do not require or use weight rewinding, while demonstrating better sparsity and generalization results than the authors present on larger models and datasets.
* The background cites none of the predominant dynamic sparse training work over the last 5 years, never mind the SOTA work from within the last year. For example, Sparse Evolutionary Training (SET), Rigging the Lottery Ticket (RigL), MEST, Gradual Magnitude Pruning . The mainly cited paper (McDanel et al) is an arxiv preprint from 2022 that appears to have not been published anywhere, and outlines a pruning (not sparse training) method.
* The paper relies on evaluation on only tiny toy datasets (MNIST, CIFAR10), and in the case of CIFAR-10, completely inappropriate and over-parameterized models for CIFAR-10 (VGG-19, ResNet-34 and ResNet50. Even in this setting the results do not achieve as high a sparsity or generalization on CIFAR-10 as demonstrated by existing DST methods (e.g. SET/RigL).
* On the only part of the paper I believe is anywhere near a novel and interesting contribution, online weight rewinding, the authors compare their online weight rewinding method - i.e. choosing weights from an iteration k - and compare those to the baselines of random initialization (k=0) and dense training (full training). A proper evaluation would compare with the exhaustive weight rewinding of LTH the authors motivate their work with. Using any weight rewinding (i.e. k>0) is going to surpass the generalization of k=0 trivially.

I can understand missing one or two citations, or coming from a different research community and being unaware of SOTA papers. This however can only be described as the willful ignorance of an entire field of research clearly demonstrated by both citing an excellent and comprehensive survey paper of the field of sparse training (Torsten Hoefler et al.) while co-opting the terminology within that survey paper, citing none of the work from it, and writing a motivation that makes no sense in context of what is written in the survey paper alone.

**Questions:**

* How does your method compare to SOTA dynamic sparse training methods such as RigL, SET and MEST?
* How does your online rewinding method compare to more appropriate baselines, including the exhaustive LTH weight rewinding, and perhaps even a random k weight rewinding point.

**Details Of Ethics Concerns:**

Even a quick google search of "Dynamic Sparse Training" clearly shows many of the works I cited above, it really makes no sense to me that a human could go through all the trouble of writing this paper while citing the best survey paper on DST/sparse training. It's either an LLM or unbelievable human ignorance... I hope this is an LLM. Furthermore the motivation just makes no sense, it's combining the weaknesses of two separate areas of research within sparse training (LTH and DST), it could only makes sense at a high level if you don't understand the literature.

---

> ### Author Response · Authors · 2025-12-03
> **Reviewer zSD9: Concerns and Responses**
>
> **Q8-Clarifying the Scope and Positioning of GDSTAR (DST vs. Rewinding)_Rev-zSD9.**
> We apologize for the ambiguity in framing GDSTAR as a “dynamic sparse training” approach. Our method is not intended to compete directly with regrowth-based DST algorithms such as RigL, SET, or MEST, which represent the state of the art in large-scale sparse training and rely on connectivity regrowth + gradient-based updates, not rewinding.
> Instead, GDSTAR focuses on a different technical question: Can the stability signals used in LTH-style rewinding be detected online and used to guide pruning during a single training run, eliminating the need for offline exhaustive rewinding?
> Thus, GDSTAR is a rewinding-based pruning framework, not a connectivity-regrowth DST pipeline. Our work sits at the intersection of weight rewinding (LTH) and pruning-based sparse training, which led us to emphasize rewinding-oriented methods rather than fully regrowth-based DST methods. It aims to (1) unify pruning and rewinding in one online schedule, (2) eliminate offline exhaustive retraining, and (3) study stability-based sparsification in a controlled environment.
> We modify Section 2  to discuss more related work.
>
> **Q9-Related Work Coverage (RigL, SET, MEST, Top-KAST, AC/DC, NeurRev, etc.)_Rev-zSD9.**
> Please see Q2 for the general related-work expansion. We modify Section 2 to discuss more about related work. The McDanel et al. method is retained as a pruning baseline because it offers a clear non-adaptive reference point in a closely related experimental regime. In the revised version, we also compare the proposed idea against random-k rewinding and LTH-based rewinding in A.7 and A.8 (Appendix) of the revised manuscript, respectively.
>
> **Q10-Ethical Concern / “Willful Ignorance” Remark_Rev-zSD9.**
> We appreciate the reviewer’s concern and want to emphasize that the incomplete citation coverage resulted from a narrow focus on rewinding-oriented methods rather than an attempt to overlook key DST work. We thank the reviewer for pointing this out. We have corrected the literature review accordingly and clarified the methodological scope to prevent misinterpretation in Section 2.
>
> **Q11-Why use CIFAR-10 with overparameterized models_Rev-zSD9.**
> This point is addressed in Q1. Briefly, overparameterized models on CIFAR-10 are standard in pruning and rewinding studies and allow us to probe sparsification dynamics in a controlled setting. At the same time, GDSTAR considers efficiency-oriented architectures such as MobileNetV2, ShuffleNetV2, and EfficientNetB0.
> We also clarify the scalability of GDSTAR and add Tiny ImageNet/ResNet-50 results in the A.6 (Appendix) of the revised version.
>
> **Q12-Baselines for rewinding: exhaustive LTH and random-k_Rev-zSD9.**
> We appreciate the reviewer’s point that comparing only to k=0 and dense training is insufficient. In the revision, we include two additional baselines: (1) a random-k rewinding baseline, where GDSTAR consistently outperforms the random selection of rewind points, see A.7 (Appendix) and (2) an LTH-based exhaustive rewinding analysis, where we compare GDSTAR to the optimal k identified by the LTH procedure, see  A.8 (Appendix). These additions provide a clearer and more comprehensive evaluation of GDSTAR’s rewinding mechanism.

---

### Official Review · Reviewer_t48f · 2025-10-31

**Soundness:** 3
**Presentation:** 2
**Contribution:** 2
**Rating:** 2
**Confidence:** 3

**Summary:**

This paper proposes GDSTAR, a Gradient-based Dynamic Sparse Training framework with Adaptive Rewinding, aiming to achieve efficient sparse training without offline retraining.
Experiments on 11 DNNs (LeNet, ResNet, VGG, EfficientNet, etc.) over MNIST and CIFAR-10 show that GDSTAR achieves up to 96% sparsity with an average 0.94% accuracy drop, outperforming prior dynamic sparse training methods like Procrustes by 0.72% on average.

**Strengths:**

1. Proposes an online adaptive rewinding mechanism using the Frobenius norm of gradients.
2. Well-motivated and mathematically clear.
3. Demonstrates performance gains.

**Weaknesses:**

1. Experiments are focused on MNIST and CIFAR-10, which are very small datasets. It would be good to have the experimental results on ImageNet.
2. The paper does not provide quantitative runtime or FLOPs comparisons.
3. The experiments only compare the proposed method with Procrustes, but no other sparse training baselines.

**Questions:**

1. Could GDSTAR be extended to ImageNet-scale models or non-CNN architectures (e.g., transformers or ViTs)?
2. How often is the optimal rewind point updated? Is frequent updating beneficial or redundant after early stabilization?
3. How much additional time or memory does online Frobenius-norm computation add compared to Procrustes or standard training?

---

> ### Author Response · Authors · 2025-12-03
> **Reviewer t48f: Concerns and Responses**
>
> **Q4-Evaluation Beyond MNIST/CIFAR-10 (Scalability to ImageNet)_Rev-t48f.**
> Please see Q1 for our general discussion on experimental scope and scalability. In brief, we use overparameterized architectures on CIFAR-10 to isolate adaptive rewinding dynamics in a controlled setting, and GDSTAR scales naturally to larger models because it only relies on gradients and does not modify the forward pass. In the revised version, we have added Tiny ImageNet experiments with ResNet-50 to partially bridge the gap to ImageNet-scale evaluation.
>
> **Q5-Runtime, FLOPs, and Memory Overhead_Rev-t48f.**
> This concern is addressed in detail in Q3. We emphasize here that GDSTAR introduces 2.53% training-time overhead, on average, and does not require additional forward/backward passes. In the revised manuscript, we provide a thorough analysis of the overhead, including training time, memory usage, and computational cost in A.9 (Appendix).
>
>
> **Q6-Comparison to Other Sparse Training Baselines_Rev-t48f.**
> As discussed in Q2, we have expanded the related work to cover recent DST and pruning methods, and we clarify that our goal is to isolate the effect of the rewinding schedule rather than to build a full DST pipeline. In the revised version, we add comparisons to random-k rewinding and LTH-based rewinding in A.7 and A.8 (Appendix) of the revised manuscript, respectively.
>
> **Q7-Frequency and Stability of Rewind-Point Updates_Rev-t48f.**
> In Figure 3, we show the plateau behavior of the stability signal. We clarify the update frequency of the rewind point in Section 3.1 and in Algorithm 4 (Appendix). GDSTAR does not evaluate the rewind criterion at every iteration. Instead, it monitors the Frobenius norm once per pruning step (i.e., once per epoch in our experiments) and updates the rewind point only when a new running minima of the stability signal is detected through a lightweight procedure.
> The computational and runtime overhead of tracking this stability signal is minimal: the running minimum is computed directly from gradients already produced during backpropagation, and the total added cost is only 2.53% of end-to-end training time. This small overhead demonstrates that calculating and updating the rewind point introduces negligible additional computation.
> Finally, in the revised version, we provide a thorough analysis of all overhead components, including runtime, memory footprint, and the additional computation required for storing and updating the rewind checkpoint, to give a complete and transparent view of the method’s efficiency.

---

### Author Response · Authors · 2025-12-03
**Common Concerns**

We sincerely appreciate the reviewer’s detailed analysis and the opportunity to clarify the scope and positioning of our work. We have revised the manuscript to (1) clarify the positioning of GDSTAR w.r.t. DST and LTH, (2) expand the related work to cover recent DST methods, and (3) add analyses of overhead, hyperparameter sensitivity, and experimental details. We address all concerns below.

**Common questions:**

**Q1-Evaluation-Large-Datasets-Models_Reviwers-t48f-zSD9-WDCn-k4Q7.**
We agree that ResNet-18/34, VGG-16/19 are overparameterized for CIFAR-10. However, this setting (Overparameterized architectures on CIFAR-10) is a standard benchmark in pruning and rewinding research. It is because our focus in this submission was to isolate and analyze adaptive rewinding dynamics in a controlled environment where the effect of rewinding is not confounded by regrowth heuristics or dataset-specific hyperparameter tuning.
At the same time, GDSTAR is not restricted to overparameterized networks. We additionally report results on efficiency-oriented architectures such as MobileNetV2, ShuffleNetV2, and EfficientNetB0. These models are not overparameterized for CIFAR-10, and we observe that GDSTAR generalizes well to them, further supporting the method’s robustness across diverse model families.
We also agree that larger-scale datasets such as ImageNet are important for demonstrating the generality of GDSTAR. GDSTAR scales naturally to large models and datasets because (1) it only requires access to gradient tensors and does not modify the forward propagation, and (2) the GDSTAR does not depend on model size; the Frobenius norm is computed from gradients that are already available during backpropagation for all kinds of architectures.
Due to computational limitations during submission, we were unable to include full ImageNet experiments. We are currently extending the implementation to ImageNet and larger architectures, and have already added results for ResNet-50 trained on Tiny ImageNet in the revised version.
We clarify this in A.6 (Appendix) of the revised paper.

**Q2-Evaluation-other sparse training baselines and cite_Reviwers-zSD9-k4Q7.**
We acknowledge that evaluating only against Procrustes was insufficient and appreciate the reviewer’s suggestion to discuss recent DST methods, such as Sparse Evolutionary Training (SET), Rigging the Lottery Ticket (RigL), MEST, Gradual Magnitude Pruning, Top-KAST, AC/DC, NeurRev, and other works [1–8]. This was an oversight, not an intentional omission.
In the revised manuscript, Section 2, we expand the related work section to clarify the connection and to position our contribution more explicitly relative to the mentioned approaches, such as dynamic sparse training with gradient-based or regrowth-based connectivity updates (e.g., SET, RigL, MEST, and [1-8]), and pruning-oriented schedules such as gradual magnitude pruning and compression methods, such as AC/DC. These methods introduce diverse mechanisms such as gradient-based regrowth, plasticity-aware connectivity updates, or alternating compression cycles.  The method of McDanel et al., which we use as a pruning baseline, is included because it provides a clear reference point for non-adaptive pruning/rewinding behavior in a similar experimental regime, even though it is a pruning method rather than a full dynamic sparse training algorithm.

Our intention is not to replicate full DST pipelines but to isolate the contribution of the rewinding schedule. Instead, GDSTAR focuses on a different technical question: Can the stability signals used in LTH-style rewinding be detected online and used to guide pruning during a single training run, eliminating the need for offline exhaustive rewinding?
We modify Section 2 to discuss more related work.

**Q3-Evaluation-Overhead-Time-Memory_Reviewers-t48f-WDCn.**
We appreciate the reviewer highlighting the importance of runtime analysis. GDSTAR is designed to incur minimal overhead because (1) it reuses the gradients already computed during backpropagation, (2) the Frobenius-norm computation reuses per-layer gradients, requiring only a reduction operation, (3) it does not require additional forward/backward passes unlike offline rewinding methods that require multiple full retraining runs, and (4) memory cost is unchanged except for storing a single candidate rewind checkpoint.
In the revised manuscript, we provide a thorough analysis of the overhead, including training time, memory usage, and computational cost in A.9 (Appendix). As an example, GDSTAR introduces 2.53% training-time overhead, on average. We clarify that real wall-clock speedups depend on hardware-level sparse kernel support, which varies across accelerators.

---

### Meta-Review · Area_Chair_VFqx · 2026-01-06

**Summary:**

The proposed method provides a unified approach to integrate pruning and adaptive rewinding directly into training of deep neural networks, reducing offline retraining costs. The method is sound and evaluated in small scale settings. The major concerns raised by the reviewers include the scale of the experiments, the reduced set of baselines, the inadequate positioning of the work wrt SotA, as well as the absence of a runtime and a memory analysis. Each of these concerns were raised by several reviewers.

**Reviewer Concerns:**

The state on the major concerns raised by the reviewers can be summarised as follows:
- Several reviewers indicated that experiments were only conducted on very small data sets, raising the question how the results translate to larger data sets and models. Moreover, the reported performance improvements suggests the contributions are of modest interest: small drop of performance compared to unpruned methods and small gains compared to what the authors considered as SotA. This suggests that SotA is already strong, leaving little room for gains (aside from runtime gains), at least in the setting considered by the authors.
- Several reviewers found that the use of overparametrised models was inappropriate. The authors adequately addressed this criticism and I concur with them that this is a common setting used in previous work. The additional results produced by the authors was also helpful.
- Several reviewers asked for a quantitative runtime analysis. This was also addressed by the authors in their rebuttal.
- The reviewers questioned the positioning of this paper compared to SotA. Both the most critical and most positive reviewer indicated that the positioning of this work wrt SotA was lacking. Reviewers also found that insufficient baselines were considered in the experiments. This was partially addressed by the authors: while this improved the positioning of the paper and offers a more objective picture, I am of the opinion that most of this material should have been part of the original submission.

In addition:
- Reviewer WDCn questioned the sensitivity to the hyperparamerter setting. The authors provided necessary clarifications in their rebuttal.
- Finally, reviewer zSD9 indicated that the novelty of this work was limited and that the online rewinding mechanism was not evaluated appropriately. This specific point was acknowledged by the authors. They provided new results in the rebuttal to address this point.

**Reviewer Scores:**

Three out of 4 reviewers voted for rejection. I would have expected the most critical reviewer to raise their score post rebuttal, after reading the clarifications of the reviewers and the additional material presented. However, this paper would have remained a borderline reject paper given the initial set of flaws identified by the reviewers (eg, positioning, small scale experiments, lacking baselines, modest novelty). Hence, I cannot recommend acceptance.

---

### Decision · Program_Chairs · 2026-01-26

Reject